# Prenatal Identification of a Novel Mutation in the *MCPH1* Gene Associated with Autosomal Recessive Primary Microcephaly (MCPH) Using Next Generation Sequencing (NGS): A Case Report and Review of the Literature

**DOI:** 10.3390/children9121879

**Published:** 2022-11-30

**Authors:** Ioannis Papoulidis, Makarios Eleftheriades, Emmanouil Manolakos, Michael B. Petersen, Simoni Marina Liappi, Anastasia Konstantinidou, Maria Papamichail, Vassilios Papadopoulos, Antonios Garas, Sotirios Sotiriou, Ioannis Papastefanou, Georgios Daskalakis, Aleksandar Ristic

**Affiliations:** 1Access to Genome P.C., Clinical Laboratory Genetics, Lampsakou 11, 11528 Thessaloniki, Greece; 2Second Department of Obstetrics and Gynaecology, Aretaieion Hospital, Medical School, National and Kapodistrian University of Athens, 112527 Athens, Greece; 3Department of Medical Genetics, University of Cagliari, Binaghi Hospital, 09124 Cagliari, Italy; 41st Department of Pathology, School of Medicine, National and Kapodistrian University of Athens, 11528 Athens, Greece; 5Postgraduate Programme “Maternal Fetal Medicine”, Medical School, National and Kapodistrian University of Athens, 11528 Athens, Greece; 6Department of Obstetrics & Gynecology, University of Patra, 26500 Patras, Greece; 7Department of Gynecology, Larissa Medical School, University of Thessaly, 38221 Larissa, Greece; 8Department of Clinical Embryology, Larissa Medical School, University of Thessaly, 41334 Larissa, Greece; 9Fetal Medicine Clinic, Monis Petraki 4, Kolonaki, 11521 Athens, Greece; 10First Department of Obstetrics and Gynaecology, “Alexandra” Maternity Hospital, Medical School, National and Kapodistrian University of Athens, 15772 Athens, Greece; 11Obstetric and Gynecological Clinic Narodni Front, 11000 Belgrade, Serbia

**Keywords:** homozygous *MCPH1* mutation, microcephalin gene, microcephaly, Next Generation Sequencing

## Abstract

Background: *MCPH1* is known as the microcephalin gene (OMIM: *607117), of which the encoding protein is a basic regulator of chromosome condensation (BCRT-BRCA1 C-terminus). The microcephalin protein is made up of three BCRT domains and conserved tandem repeats of interacting phospho-peptides. There is a strong connection between mutations of the *MCPH1* gene and reduced brain growth. Specifically, individuals with such mutations have underdeveloped brains, varying levels of mental retardation, delayed speech and poor language skills. Methods: In this article, a family with two affected fetuses presenting a mutation of the *MCPH1* gene is reported. During the first trimester ultrasound of the second pregnancy, the measure of nuchal translucency was increased (NT = 3.1 mm) and, therefore, the risk for chromosomal abnormalities was high. Chorionic villi sampling (CVS) was then performed. Afterwards, fetal karyotyping and Next Generation Sequencing were carried out. Afterwards, NGS was also performed in a preserved sample of the first fetus which was terminated due to microcephaly. Results: In this case, the fetuses had a novel homozygous mutation of the *MCPH1* gene (c.348del). Their parents were heterozygous for the mutation. The fetuses showed severe microcephaly. Because of the splice sites in introns, this mutation causes the forming of dysfunctional proteins which lack crucial domains of the C-terminus. Conclusion: Our findings portray an association between the new *MCPH1* mutation (c.348del) and the clinical features of autosomal recessive primary microcephaly (MCPH), contributing to a broader spectrum related to these pathologies. To our knowledge, this is the first prenatal diagnosis of MCPH due to a novel *MCPH1* mutation.

## 1. Introduction

Autosomal recessive primary microcephaly (MCPH) is a heterogenous disorder of neurogenic brain development [1]. It is characterized by decreased head circumference at child birth with no severe brain anatomical abnormalities and varying mental retardation [2,3]. There are two categories of microcephaly: the first is diagnosed shortly after birth and is called primary microcephaly (MCPH), while the second develops later in life and is called secondary microcephaly. The main difference between primary and secondary microcephaly is that MCPH in the vast majority of the cases is a static irregularity of development, while secondary microcephaly is a dynamic neurodegenerative disorder [2,3].

Whole genome sequencing (WGS) or whole exome sequencing (WES) studies have identified 30 OMIM genes associated with MCPH (OMIM date, February 2022), and those are: *WDR62* (*613583), *CDK5RAP2* (*608201), *CASC5* (*609173), *ASPM* (*605481), *CENPJ* (*609279), *STIL* (*181590), *CEP135* (*611423), *CEP152* (*613529), *ZNF335* (*610827), *PHC1* (*602978), *CDK6* (*603368), *CENPE* (*117143), *PLK4* (*605031), *TUBGCP6* (*610053), *NDE1* (*609449), *KLN1* (*609173), *SASS6* (*609321), *MFSD2A* (*614397), *ANKLE2* (*616062), *CIT* (*605629), *KIF14* (*611279), *NCAPD2* (*615638), *NSAPD3* (*609276), *NSAPH* (*602332), *NUP37* (*609264), *MFSDA* (*614397), *MAP11* (*618350), *LMNB1* (*150340), *LMNB2* (*150341) and *RRP7A* (*619449) [1,2]. As the *MCPH1* gene (*607117) had already been associated with MCPH, it is concluded that 31 genes in total are associated with MCPH [3,4]. All the genes mentioned above encode proteins which have a crucial role in the neurogenic programming as they control the cell-cycle checkpoints and cell signaling.

The *MCPH1*/microcephalin gene (*607117) is located in chromosome 8p23. Cellular responses caused by DNA damage and chromosome condensation are dependent on microcephalin, the encoded protein [4]. This protein is involved in G2/M checkpoint arrest, maintaining the inhibitory phosphorylation of cyclin-dependent kinase 1. The ability to maintain normal brain size is lost when microcephalin is absent because it causes premature mitotic entry of neuroprogenitor cells [4,5,6,7].

In this study, we report a family with two affected fetuses which were both homozygous for an *MCPH1* gene mutation. Both the fetuses were growth-restricted with microcephaly, while the first fetus also presented partial agenesis of the corpus callosum. To our knowledge, this the first case of a prenatal diagnosis of MCPH due to a novel *MCPH1* mutation.

## 2. Materials and Methods

### 2.1. Clinical Report

A 30-year-old woman was referred to our center during her second pregnancy at 24 weeks of gestation, complicated with microcephaly. Concerning the medical history of the parents, they were healthy and they had normal mental development.

In their first pregnancy, during the ultrasound examination (U/S) of the first trimester, nuchal translucency measurement was within normal limits (NT = 1.3 mm), indicating low risk for Down syndrome (1/9052). The second trimester ultrasound examination (U/S) at 24 weeks of gestation revealed Fetal Growth Restriction (FGR), microcephaly and partial agenesis of the corpus callosum, dilatation of the third ventricle, right ureter dilatation and the presence of a simple unilocular cyst at the lower lobe of the right kidney. After counselling, the parents opted for fetal karyotyping, and amniocentesis was performed. Standard G-banding karyotyping and chromosomal microarray analysis were normal. After genetic counselling the pregnancy was terminated at 26 weeks of gestation, and a male fetus was delivered. The parents decided against autopsy, and no further analysis was performed at this point. The parents were of Greek origin, healthy and nonconsanguineous.

During the second pregnancy, at the first trimester ultrasound examination (U/S) at 12 weeks of gestation, the risk for chromosomal abnormalities was increased, due to an increased nuchal translucency measurement (NT = 3.1 mm). After counseling, the parents opted for fetal karyotyping, and chorionic villus sampling (CVS) was then performed. Standard G-banding karyotyping revealed a normal karyotype, and chromosomal microarray analysis did not detect any pathogenic copy number variations. During the second trimester of pregnancy, ultrasound assessment showed severe microcephaly and FGR (Fetal Growth Restriction). Following pregnancy termination, fetal autopsy, standard G-banding karyotyping, chromosomal microarray and NGS analysis were performed.

### 2.2. Molecular Karyotyping

High-resolution molecular karyotyping was performed with an aCGH platform of 60,000 oligonucleotides (Agilent technologies, California, USA). Briefly, DNA was extracted from the chorionic villus sample with Promega Maxwell 16, and it was hybridized with the human reference DNA of the same gender (Promega Biotech). The statistical test used as a parameter to estimate the number of copies was ADM-2 (provided by the DNA analytics software, Agilent Technologies) with a window of 0.5 Mb and a threshold of 6. Only those copy number changes that affected at least 5 consecutive probes with identically oriented change were considered as real copy number variations. Consequently, for most of the genome, the average resolution of this analysis was 200 kb.

### 2.3. Next Generation Sequencing (NGS)

DNA isolation from the fetus and the parents was performed. Following, oligonucleotide-based target capture analysis and nucleotide sequencing were performed using TruSight One kit (Illumina) and Next Generation Sequencing (Illumina NextSeq), respectively, examining 99 disease-causing genes that have been associated with the clinical features of the fetus (Table 1). Although a large set of the commonest genes (including 99 genes) that have been associated with MCPH were analyzed, a small number of other genes that correlate with the fetus’ phenotype were not included in the analysis (*PHC1*, *SASS6*, *NUP37*, *MAP11*, *LMNB1,* etc.). Therefore, additional mutations that might be implicated in the phenotype cannot be excluded. The software SOPHIA DDM^®^ (SOPHIA GENETICS) was used for the data analysis [reference genome UCSC hg19 and reference database Human Gene Mutation Database (HGMD v.2017.1)].

## 3. Results

Standard G-banding karyotyping and high-resolution molecular karyotyping revealed a normal chromosome chart of the second fetus. NGS analysis showed an inherited homozygosity for a mutation of the *MCPH1* gene (c.348del) which has not been reported in polymorphism databases (ExaC or exome variant server (EVS)) (shown in Figure 1). The results were confirmed by Sanger sequencing in this fetus, and targeted sequencing on the parents showed that they were both heterozygous for the detected mutation (shown in Figure 2). Afterwards, targeted sequencing of the DNA of the first fetus was performed, also revealing homozygosity for the same mutation.

The results were confirmed by Sanger sequencing in this fetus, revealing mutation c.348del in the *MCPH1* gene detected in homozygosity, and targeted sequencing of the parents showed that they were both heterozygous for the detected mutation (shown in Figure 2A,B). Afterwards, targeted sequencing of the DNA of the first fetus was performed, also revealing homozygosity for the same mutation c.348del in the MCPH1 gene.

### Pathologic Results

Fetal necrotomy of fetus number two revealed mild symmetric growth retardation, mild dysmorphic craniofacial features, microcephaly and partial agenesis of the corpus callosum without histologic cerebral dysplasia. The placental weight was normal (SD: −1.2), but histologic examination revealed extended inflammatory lesions of maternal origin dating earlier in the pregnancy (acute chorionitis and villitis, karyorrhexis, fibrin deposition) and multifocal chronic villitis of unknown origin. There were no indications of a specific syndromic microcephaly or partial corpus callosum agenesis. The possibility of “microcephaly-corpus callosum agenesis syndrome” (ORPHA NB:171703, no OMIM number exists), or other cases of isolated genetic microcephaly, could not be excluded. The inflammatory lesions of the placenta could imply the possibility of an intrauterine teratogenic effect due to an infectious agent, although no analogous histologic findings of the cerebrum were observed.

## 4. Discussion

The mode of inheritance of the genetically heterogenous disorder primary microcephaly (MCPH) is autosomal recessive. MCPH is characterized by a congenital small cranium with a reduced occipito-frontal head diameter (OFD) of more than two standard deviations (SD) below the mean for ethnicity, age and sex (heavy microcephaly OFD < 3 SD). Other common clinical findings reported among patients are a simplified gyral pattern, mild to moderate intellectual disability, polymicrogyria, periventricular neuronal heterotopias, hyperactivity and attention deficit disorder, speech delay, focal or generalized seizures, aggressiveness, delay of developmental milestones, facial dysmorphism, hereditary hearing loss and pyramidal signs [8,9,10].

The incidence of MCPH is about 1:30,000–1:250,000 live births [9], and more than 300 families worldwide have been reported to manifest MCPH. The disease is less common in Caucasian than in Arab and Asian populations, where consanguineous marriages are usual [11].

The *MCPH1*/microcephalin gene contains 241,905 bp. A total of 835 amino acids are encoded by 14 exons which are in the 8032 bp open reading frame; until today, three isoforms are known [4]. The microcephalin protein is a pleiotropic factor and plays an important role in neurogenesis. It regulates the division of neuroprogenitor cells preventing them to exhaust, regulates the telomere integrity, is involved in G2/M checkpoint arrest maintaining the inhibitory phosphorylation of cyclin-dependent kinase 1 [5,6,7], operates as a tumor suppressor in many human cancers and regulates the cerebral cortex size as well as the development of the brain [12,13,14].

Two types of mutations in the *MCPH1* have been described. Mild microcephaly is manifested in patients who were identified with a missense mutation [15,16,17,18,19,20]. Furthermore, there are reported deletions in *MCPH1*. An Iranian family with mild microcephaly and intellectual disability was identified with a deletion of the first six exons of the gene; at least 10% to 15% of the cells had early chromosome condensation in this family [17]. Moreover, an Asian–Indian patient was identified with a deletion of the first 11 exons [18]. Recently, Naseer et al. [21] studied four generations of a Saudi family in which primary microcephaly was common. They identified two missense variants (c.982G > A and c.1273T > A) in a heterozygous state in exon eight of the *MCPH1* gene. The affected individuals presented severe microcephaly (−5 SD and −6 SD below for the same gender and age) with typical dysmorphism of the disorder, such as the sloppy forehead. They were mentally retarded and their emotional and verbal intelligence were not appropriate for their age. They had no neurological impairments, such as increased reflexes or seizures. One other equally interesting study is the report of Pavone et al. [22]. In their report, the authors presented two affected female twins with heterozygous missense mutations in the *MCPH1* gene (c.2180C > T). The girls had microcephaly (−3 SD below for the same gender and age) with no facial dysmorphism or neurological impairments. They attended a normal elementary school, but had learning difficulties requiring special support. Magnetic resonance imaging (MRI) in one of the patients, performed at the age of 2, showed microcephaly with a small brain, affecting particularly the frontal lobe, while the cortex and corpus callosum were normal. However, in this study, heterozygosity for one missense mutation is reported, and the authors state the hypothesis that other factors in association with the variant, including epigenetic events, may have a role in causing the impaired cerebral development manifested by the twins.

In this report, we present two consecutive pregnancies complicated by severe microcephaly, caused by a homozygous mutation of the *MCPH1* gene. Both parents of the embryo carried the c.348del and were heterozygous. This mutation leads to the replacement of the phenylalanine amino acid for the leucine amino acid at position 116 of the protein, altering the reading frame which causes an early stop codon at position 145 of the protein, leading to protein forms which lose their function. The *MCPH1* mutation (c.348del) has been revealed by our findings and appears to result in microcephaly clinical features contributing to the *MCPH1* spectrum’s expansion associated with these pathologies. The mutations of the *MCPH1* gene were those which were firstly identified in MCPH [9]. Furthermore, there few mutations in the *MCPH1* gene have been reported, and to our knowledge, there are no reports of a prenatal diagnosis of an *MCPH1* gene.

## 5. Conclusions

The diagnosis of primary microcephaly was based on the identification of an *MCPH1* mutation using molecular techniques. Despite the fact that we found a novel mutation in homozygosity associated with fetal abnormalities, additional studies are necessary to verify the pathogenicity of this mutation and its definite association with fetal microcephaly. We present this case of the novel mutation c.348del to share information and expand our limited knowledge on the spectrum of disease linked to *MCPH1* gene mutations and its prenatal diagnosis.

## Figures and Tables

**Figure 1 children-09-01879-f001:**
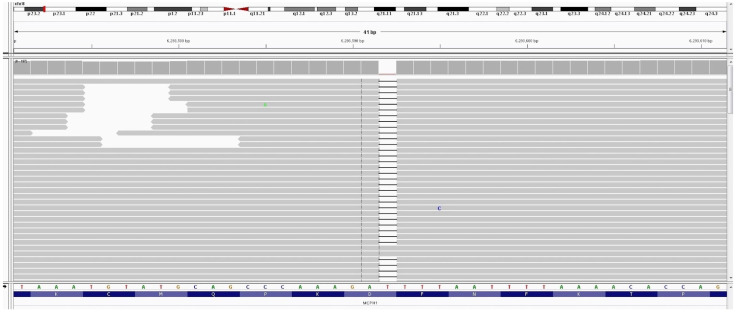
IGV screenshot for mutations in the *MCPH1* gene. The mutation c.348del in the *MCPH1*.

**Figure 2 children-09-01879-f002:**
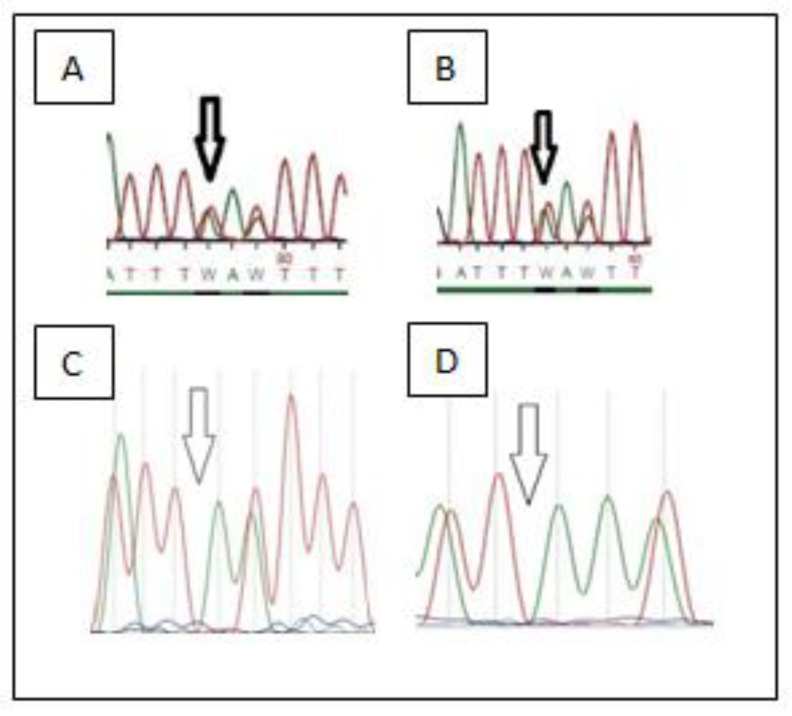
(**A**) Sanger sequencing data of the mutation c.348del in the MCPH1 gene in heterozygous state (mother). (**B**) Sanger sequencing data of the mutation c.348del in the MCPH1 gene in heterozygous state (father). (**C**) Mutation c.348del in the MCPH1 gene detected in homozygosity in fetus 1. (**D**) Mutation c.348del in the MCPH1 gene detected in homozygosity in fetus 2.

**Table 1 children-09-01879-t001:** A total of 99 disease-causing genes that were related with the clinical features of the fetus.

*AP3B2*, *AP4B1*, *AP4E1*, *AP4M1*, *APEX1*, *ARFGEF2*, *ASNS*, *ASPM*, *ATR*, *ATRX*, *BMPR1A*, *BRWD3*, *C2*, *CDK5RAP2*, *CDON*, *CDT1*, *CENPJ*, *CEP152*, *COL2A1*, *CYP21A2*, *CZ1P-ASNS*, *DICER1*, *DPP6*, *EFTUD2*, *EHMT2*, *EMG1*, *F10*, *F2*, *F5*, *FAM20A*, *FAM20C*, *FLNA*, *FLT1*, *FN3K*, *GLI2*, *GNAS*, *HUWE1*, *HYMAI*, *IGF1R*, *KARS*, *KNL1*, *LIG4*, *LMNB2*, *LOC100287042*, *LOC102724058*, *MCM5*, *MCPH1*, *MCPH1-AS1*, *MED17*, *MMP2*, *MRE11*, *MTHFR*, *MTNR1A*, *MYH11*, *NBN*, *NCAPD2*, *NDE1*, *NELFE*, *NPC1*, *ORC4*, *ORC6*, *OSGEP*, *PACERR*, *PCNT*, *PEX2*, *PHB2*, *PLAGL1*, *PRKAR1A*, *PRKDC*, *PTGS1*, *PTGS2*, *PTPRJ*, *REV3L*, *RPTOR*, *SCAP*, *SCARNA10*, *SCN10A*, *SCN1A*, *SERPINA3*, *SKIV2L*, *SLC16A2*, *SLC25A19*, *STIL*, *STK4*, *TBCD*, *TCOF1*, *TERT*, *TP53*, *TUBA1A*, *TUBGCP6*, *VPS13B*, *VPS35*, *VRK1*, *WDR62*, *WDR81*, *XRN1*, *ZNF335*, *ZNF592*, *ZNF750*

## Data Availability

Not applicable.

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
