# Peer review of "Prenatal Identification of a Novel Mutation in the MCPH1 Gene Associated with Autosomal Recessive Primary Microcephaly (MCPH) Using Next Generation Sequencing (NGS): A Case Report and Review of the Literature"

_children, 2022, doi:10.3390/children9121879_

Round 1
Reviewer 1 Report (New Reviewer)
Summary: This a case report on Autosomal recessive primary microcephaly, reporting a new mutation associated with the disease. In addition the article provides a literature review relevant to the case adding to the discussion and the novelty of the author’s findings. The article is well written and teh title adequately describes the presented case and findings.
General concept comments:
Article: This case report has adequate methodological accuracy, well supporting its findings.
Minor Comments:
A pedigree with the ancestry of the parents would be potentially useful to the reader and the study, to identify other possibly affected subjects.
Line 61-62, can authors provide a reference? Literature indicates that MCPH can often present with global developmental delay and gray matter heterotopias. In addition this statement partially contradicts the first paragraph of the article’s discussion (https://rarediseases.info.nih.gov/diseases/12117/autosomal-recessive-primary-microcephaly)
Author Response
Dear Reviewers,
First of all, we would like to thank you for our to the point comments, that improved radically our manuscript.
Concerning the comments of reviewer No1:
We added the missing references on paragraph 1. We deleted entirely the sentence on line 60-61, as it was conflicted with the 1st paragraph of “Discussion”. Concerning the ancestry of the parents we state in the manuscript that they are healthy, nonconsanguineous and Greek origin. Unfortunately, we do not have any data from the grandparents and other family members.
Reviewer 2 Report (New Reviewer)
The manuscript entitled "Prenatal identification of a novel Mutation in the MCPH1 Gene associated with autosomal recessive primary microcephaly (MCPH) using Next Generation Sequencing (NGS): A Case Report and Review of Literature" submitted by Papoulidis et al describes two cases with patients with autosomal recessive primary microcephaly (MCPH). The manuscript contains a clinical report on two pregnancies with results from ultrasound examination, amniocentesis, fetal karyotyping, chromosomal microarray analysis and chorionic villus sampling. DNA was isolated from fetus and parents and oligonucleotide-based target capture analysis and DNA sequencing was performed using TruSight One Kit and NGS, respectively. Clinical data for both fetuses from second trimester reveled clear signs of severe microcephaly. Subsequent DNA sequencing confirmed this diagnosis revealing a novel homozygous in the MCPH1 gene (c.348del). The parents were heterozygous for this deletion mutation, which results in a dysfunctional MCPH1 protein. The authors present an interesting study which contributes to the understanding of genetic causes for diseases, in this case microcephaly. The methods used in this study seem to be of sound quality and up to date. The data presentation (graphics) is not of the best quality and needs to be improved. Also, the discussion could benefit with some condensing and check for many repeated sentences. For instance, some sentences are redundant, and found both in the Introduction and in the Discussion. Finally, there is some confusion about the presentation of references.
Major points:
Minor points:
Data presentation in Fig. 2 and Fig.3. The quality of the sequence data is poor with double signals in many nucleotide positions in the electropherograms for the parents! Suggestion: Fig.2 and Fig.3 could be assembled in one figure with four panels - one for each parent and one for each fetus.
Question: Comments on the publication from Pavone et al. [20]. How can the female twins display disease phenotype with heterozygous mutations in the MCPH1 gene if it is classified as autosomal recessive primary microcephaly.
Keywords should be ordered alphabetically.
Typo: line 127: been.
Line 196: do not start a sentence with a number.
Confusion with the references cited in the text: Cannot find [7] and [10]. Why is [19] before [19] and [20]?
Author Response
Dear Reviewers,
First of all, we would like to thank you for our to the point comments, that improved radically our manuscript.
Concerning the comments of reviewer No2:
We have corrected the vocabulary mistakes and the incomprehensible sentences have been rewritten. We have also corrected the references presentation.
For the comment: “Data presentation in Fig. 2 and Fig.3. The quality of the sequence data is poor with double signals in many nucleotide positions in the electropherograms for the parents”, we note that the data are from a Sanger analysis and the mutations are shown. For a better image with less noise, the mutation’s point have been highlighted. We have also improved the graphics presentation.
Following the suggestion for Fig.2 and Fig.3 that could be assembled in one figure with four panels - one for each parent and one for each fetus, it has been assembled in one image with A, B, C and D parts.
This manuscript is a resubmission of an earlier submission. The following is a list of the peer review reports and author responses from that submission.
Round 1
Reviewer 1 Report
In this manuscript Papoulidis et al. present a case report and, according to their title, review of literature. Below there is a list of major points that, in my opinion, need to be addressed.
- The introduction is not up-to-date; for instance the authors state “Whole genome sequencing (WGS) or whole exome sequencing (WES) studies identified 17 OMIM genes associated with MCPH and those are” (lines 61-62) and “it is concluded that 18 genes in total are associated with MCPH” (lines 67-68). In fact more than 25 genes have been associated with MCPH so far, so the introduction needs rewriting (for instance see a review by Jean et al. doi: 10.3389/fneur.2020.570830). This is a problem throughout the paper, papers/review articles of the last years have not been included.
- ‘In this study, we report a case of a fetus which was homozygous for the MCPH1 gene (c.348del) fetus and was diagnosed at 22 weeks of gestation’(lines 75-76). In fact they report a novel mutation for the MCPH1 gene that the detected in 2 embryos (line 84-104, 133-134).
- The authors chose a set of 99 genes for NGS, however from this set several genes than have been associated with MCPH were not included (for instance PHC1,SASS6,NUP37,MAP11, LMNB1 etc). Sequencing of these genes is important to exclude that additional mutations are also present and probably implicated in the phenotype (although, as new genes implicated in MCPH1 are still characterised this possibility cannot be excluded 100%). However, as this is a novel mutation, all genes known so far must be analysed, therefore the authors need to do also an NGS of the genes that have not been included in their initial set.
- The results can be better presented - the Figures are in low resolution, the mutation site is not indicated.
- There is a problem with the numbering, Section 3 is followed by Section 2.4!
- The Discussion needs rewriting taking in consideration literature after 2017.
- Finally a minor point- the style of the references is not the same for instance compare
Pfau RB, Thrush DL, Hamelberg E, Bartholomew D, Botes S, Pastore M, Tan C, del Gaudio D, Gasti-er-Foster JM, Astbury C. MCPH1 deletion in a newborn with severe microcephaly and premature chromosome condensation. Eur J Med Genet. 2013 Nov;56(11):609-13
with
Darvish H, Esmaeeli-Nieh S, Monajemi GB, Mohseni M, Ghasemi- Firouzabadi S, Abedini SS, Bahman I, Jamali P, Azimi S, Mojahedi F, Dehghan A, Shafeghati Y, Jankhah A, Falah M, Soltani , Banavandi MJ, Ghani M, Garshasbi M, Rakhshani F, Naghavi A, Tzschach A, Neitzel H, Ropers HH, Kuss AW, Behjati F, Kahrizi K & Najmabadi H (2010). A clinical and molecular genetic study of 112 Iranian fami-lies with primary microcephaly. Journal of Medical Genetics 47(12), 823–828.
Author Response
Dear reviewer,
thank you for your to the point comments regarding our article.
- We updated the manuscript with the new genes that have been accosiated with microcephaly.
- We corrected the information in Abstract and Introduction that in thiw article we report a family with two affected fetuses.
- Unfortunately, at this stage we cannot perform further genetic analysis since the parents do not consent for that. We will add in our manuscript as a limitation that "Although a large set of the commonest genes (99) were analysed several genes than have been associated with MCPH were not included ( PHC1,SASS6,NUP37,MAP11, LMNB1 etc) thus additional mutations that might be implicated in the phenotype cannot be excluded".
- We fixed the resolution of the figures and we moted the mutation found
- We corrected the numbering. We replaced the 2.4 with 3.1
- We added to the discussion 4 more articles from 2018 to 2021.
- We corrected this reference.
Thank you again for your comments.
Yours sincerely,
Maria Papamichail
Reviewer 2 Report
The manuscript by Papoulidis et al. reports novel a Mutation in the MCPH1 Gene associated with autosomalrecessive primary microcephaly (MCPH) using Next Generation Sequencing (NGS). the clinical reports present genetic and clinical data of the case and family. The figures are very low quality and in general I sould suggest to revise the manuscript for presentation (there are highlighted bits, progressive numbers randomly ordered). In the abstract it is stated "Its parents were recessive heterozygous for (c.348del),
displayed severe microcephaly". If the parents presented microcephaly and/or other clinical signs this should be added in the results section. Please also use appropriate nomenclature for genes (italics etc.)
Author Response
Dear reviewer,
thank you very much for your to the point comments regarding our article.
We fixed the quality of the figures and we noted the mutation found and the form that genes appear throughout the article. Also, we added the information that, despite the fact that the parents of the fetus are heterozygous for the mutation on the MCPH1 gene, they are not mentally retarded neither they have microcephaly.
We look forward to hear from you.
Yours sincerely,
Maria Papamichail
Round 2
Reviewer 1 Report
My opinion is that he revised version of the manuscript is still not suitable for publication.
Regarding my previous comments:
- The introduction has been edited to some extend but not sufficiently, the discussion has been amended also to some extend.
- lines 143-146 :"The results were confirmed by Sanger sequencing in this foetus and targeted sequencing on the parents showed that they were both heterozygous for the detected mutation (shown in Fig. 2).Afterwards, targeted sequencing of the DNA of the first foetus was performed, revealing also homozygozity for the same mutation". The results of the Sanger sequencing of the foetuses are not shown- the IGV screenshot for foetus 2 has low resolution -IGV data for foetus 1 are not shown. Essentially no data for the foetuses are presented.
Author Response
We thank the reviewer for their comments.The results were confirmed by Sanger sequencing in this fetus revealing mutation c.348del in MCPH1 gene detected in homozygosity and targeted sequencing on the parents showed that they were both heterozygous for the detected mutation (shown in Fig. 2).
Afterwards, targeted sequencing of the DNA of the first foetus was performed, revealing also homozygozity for the same mutation c.348del in MCPH1 gene.
Reviewer 2 Report
revision addressed previous concerns.
Author Response
Dear Reviewer,
Thank you for your to your to the point comments, radically improve our manuscript.
Yours sincerely,
Maria Papamichail